# Discovery of (5-Phenylfuran-2-yl)methanamine Derivatives as New Human Sirtuin 2 Inhibitors

**DOI:** 10.3390/molecules24152724

**Published:** 2019-07-26

**Authors:** Lijiao Wang, Chao Li, Wei Chen, Chen Song, Xing Zhang, Fan Yang, Chen Wang, Yuanyuan Zhang, Shan Qian, Zhouyu Wang, Lingling Yang

**Affiliations:** 1College of Food and Bioengineering, Xihua University, Sichuan 610039, China; 2College of Science, Xihua University, Sichuan 610039, China

**Keywords:** histone deacetylases, sirtuins, SIRT2, SAR studies, molecular docking

## Abstract

Human sirtuin 2 (SIRT2), a member of the sirtuin family, has been considered as a promising drug target in cancer, neurodegenerative diseases, type II diabetes, and bacterial infections. Thus, SIRT2 inhibitors have been involved in effective treatment strategies for related diseases. Using previously established fluorescence-based assays for SIRT2 activity tests, the authors screened their in-house database and identified a compound, 4-(5-((3-(quinolin-5-yl)ureido)methyl)furan-2-yl)benzoic acid (**20**), which displayed 63 ± 5% and 35 ± 3% inhibition against SIRT2 at 100 μM and 10 μM, respectively. The structure-activity relationship (SAR) analyses of a series of synthesized (5-phenylfuran-2-yl)methanamine derivatives led to the identification of a potent compound **25** with an IC_50_ value of 2.47 μM, which is more potent than AGK2 (IC_50_ = 17.75 μM). Meanwhile, **25** likely possesses better water solubility (cLogP = 1.63 and cLogS = −3.63). Finally, the molecular docking analyses indicated that **25** fitted well with the induced hydrophobic pocket of SIRT2.

## 1. Introduction

Histone deacetylases (HDACs) are enzymes that catalyze the removal of acyl groups from ε-*N*-acyl-lysine amino groups on histones and non-histone substrates. These have been identified and grouped into four classes [1,2,3]: Classes I, II, and IV HDACs are Zn^2+^-dependent metalloproteases; class III HDACs, namely sirtuins (SIRTs), use NAD^+^ as a cofactor for catalysis [4,5,6]. There are seven isotypes of sirtuins (SIRT1–7), which differ in their catalytic activity and subcellular localization [7]. The isotype SIRT2, which is located in both cytoplasm and nucleus [8], mainly catalyzes deacetylation and defatty-acylation for a variety of protein substrates, including histones H3 and H4 [9,10], and nonhistone proteins α-tubulin [11], p53 [12], Foxo1 [13], p300 [14], NFκB [15], PAR3 and PRLR [16]. Thus, SIRT2 has been shown to be involved in cell cycle regulation [11,17,18], autophagy [19], peripheral myelination [20], and immune and inflammatory responses [21,22,23]. Recently, many studies revealed that the dysregulation of SIRT2 activity is a key factor contributing to the pathogenesis of cancer [24], neurodegenerative diseases [25,26], type II diabetes [27], and bacterial infections [21,23], which makes SIRT2 a promising target for pharmaceutical intervention.

To date, except for some substrate analogues [7], a number of small molecule inhibitors targeting SIRT2 have been reported. The representative inhibitors are shown in Figure 1: The moderate potency or non-specific inhibitors Sirtinol (46 μM) [28], EX-527 (46 μM) [29,30], AGK2 (3.5 μM) [31,32], AEM1(18.5 μM) [33], AK-7 (15.5 μM) [34], and AC-93253 (6 μM) [35], the highly potent but unselective inhibitors VII (0.048 μM) and VIII (0.001 μM) [36], and the potent and highly isotype-selective SIRT2 inhibitor SirReal2 (0.4 μM) [23,37]. However, there remains a shortage of novel SIRT2 inhibitors as lead candidates for drug discovery and development.

The authors previously established a fluorescence-based method for SIRT2 inhibition tests [38,39,40], and identified a series of *N*-(3-(phenoxymethyl)phenyl)acetamide derivatives as highly selective SIRT2 inhibitors [38,41], some of which showed inhibitory activities against SIRT2 highly-expressed human breast cancer cells and non-small cell lung cancer cells. Recently, the in-house compound collection using the fluorescence-based method was screened, and a new compound was identified, 4-(5-((3-(quinolin-5-yl)ureido)methyl)furan-2-yl)benzoic acid (**20**, Figure 2), which displayed 63 ± 5% and 35 ± 3% inhibition against SIRT2 at 100 μM and 10 μM, respectively (Table 1). The scaffold of compound **20** is novel for SIRT2 inhibitors, and **20** has a relatively low molecular weight (387 Da) with moderate physicochemical properties (cLogP = 3.05, cLogS = −4.04). Thus, in this study, the authors used **20** as a starting point for further structural modifications (Linker, A, B, Figure 2) to improve the inhibitory potency against SIRT2.

## 2. Results and Discussion

### 2.1. Chemistry

This study synthesized a series of (5-phenylfuran-2-yl)methanamine derivatives using the synthetic routes outlined in Scheme 1, Scheme 2 and Scheme 3. Firstly, urea-based compounds **11**–**19** were acquired through the condensation reaction between the key intermediate **5a**–**5i** with aromatic-amine compounds **6**–**10** in the presence of triphosgene, in 82–93% yields (Scheme 1). The intermediates **5a**–**5i** were obtained by using Suzuki cross-coupling reaction between commercially available substituted iodobenzenes **1a**–**1i** with (5-formylfuran-2-yl)boronic acid (**2**), respectively. Then, the condensation reaction and reduction reaction were performed in sequence to produce the intermediates **5a**–**5i**. The carboxylic acid compounds **20**–**26** were subsequently produced through the hydrolysis reaction from the corresponding esters.

Next, the desired target compound **30**, a hydroxamic acid derivative, was prepared by a three-step sequence starting from the synthesized intermediate **4a** (Scheme 2). Sodium cyanoborohydride (NaBH_3_CN)-mediated reduction reaction was firstly performed to reduce the aldoxime group of intermediate **4a** to the hydroxylamine of intermediate **27** (54% yield), followed by condensation with 2-phenylacetyl chloride in the presence of NaHCO_3_ to give the compound **29**. Further, hydrolysis of compound **29** using 3.0 equiv NaOH led to the white solid target compound **30**. The synthesis of target compounds **32**–**37** are also depicted in Scheme 2. The reactions of commercially available amines (aniline, phenylmethanamine, and pyridin-3-ylmethanamine) or hydrazide (nicotinohydrazide) with intermediates **3a** or **3i** in the presence of hantzschester (1.2 equiv), catalytic amount of molecular sieve and trifluoroacetic acid, resulted in the reductive amination products **31**–**34**. The resulting compounds **31–33** were subsequently hydrolyzed to give the desired compounds **35**–**37** in high yields.

Finally, Scheme 3 presents the synthetic routes for compounds **39** and **43**–**52**, which contain a sulfonamide or amide linker. For sulfonamide linker compound **39**, intermediate **5a** was used to react with benzenesulfonyl chloride in the presence of Et_3_N at room temperature, and the resulting compound **38** underwent a hydrolysis reaction to give the desired target compound **39**, in 80% yield for two steps. The synthetic access to structurally diverse amide linker compounds **41**–**48** was achieved using a condensation reaction of carboxylic acid (**40**) with amine (**5a**, **5c**–**5f**) in the presence of 1-hydroxybenzotriazole (HOBT), 1-(3-(dimethylamino)propyl)-3-ethylcarbodiimide hydrochloride (EDCI), and N,N-diisopropylethylamine (DIPEA). The resulting ester-contained compounds **41**, **42**, **46** and **47** were subjected to hydrolyzation to afford the target compounds **49**–**52** in good yields.

### 2.2. SAR Studies with SIRT2

The enzyme activity assays were performed using a fluorogenic-based method [38,39,40], and Ac-Glu-Thr-Asp-Lys(Dec)-AMC, termed p2270, was used as the substrate. The SAR studies with all of the synthesized (5-phenylfuran-2-yl)methanamine derivatives (Table 1 and Table 2) were carried out. The compounds bearing various linkers or different substituents (A moiety) at 3- or 4-position of the phenyl of (5-phenylfuran-2-yl)methanamine scaffold (Table 1) were firstly investigated. Compared with the hit compound **20**, compounds **12** and **21**, containing a urea as linker, showed comparable or slightly lower SIRT2 inhibitory activities at 100 μM or 10 μM;.Carboxyl acid which contained compounds **20** and **21**, appeared to have better clogP and clogS properties than **12** (with clogP of 5.14 and clogS of −4.43). Compound **22** (23 ± 3%), bearing a thiourea linker, displayed lower inhibitory activity to SIRT2 than the corresponding compound **21** (33 ± 3%) at 10μM. Further comparison of the different linkers, including hydroxamic acid (**30**), secondary amine (**35**, **36**), sulfonamide (**39**) and amide (**49**, **50**) revealed that urea linker derivatives were likely to have more potent SIRT2 inhibition than other linker derivatives. The additional compounds with the 4-ethyl formate (**32**), 4-methyl (**43**), 4-methoxy group (**44**) replaced the 4-carboxyl of the phenyl of (5-phenylfuran-2-yl)methanamine scaffold or changed to 3-position substituents (**45**–**47**, **51**, and **52**) did not show improved inhibitory activity against SIRT2. These results indicate that the urea linker and 4-carboxyl of the phenyl of (5-phenylfuran-2-yl)methanamine scaffold may be beneficial to fit with the binding pocket of SIRT2.

The authors next synthesized compounds **23**–**26**, which contain a urea linker and 4-carboxyl of the phenyl of (5-phenylfuran-2-yl)methanamine. The tested inhibitory activities and calculated clogP and clogS values are shown in Table 2. Compounds **23**, **24**, and **26** appear to have moderate physiochemical properties, but compound **25**, with the pyridine moiety, likely possesses better water solubility (cLogP = 1.63 and cLogS = −3.63). Notably, compound **25** (99 ± 2% @ 100 μM, 90 ± 3% @ 10 μM) shows potent inhibition against SIRT2, which is substantially more potent than the structurally similar compound AGK2 (80 ± 6% @ 100 μM, 30 ± 5% @ 10 μM). Considering the fact that the introduction of pyridine at the skeleton has improved SIRT2 inhibition, a series of pyridine-containing (5-phenylfuran-2-yl)methanamine derivatives (**17**, **18**, **33**, **34**, **37** and **48**) were further synthesized. Comparing with **25**, only compounds **17** (50 ± 4% @ 100 μM, 37 ± 3% @ 10 μM) and **18** (40 ± 5% @ 100 μM, 23 ± 2% @ 10 μM), which both contained a urea linker, displayed low inhibition to SIRT2, whereas compounds **33**, **34**, **37** and **48** had almost no SIRT2 inhibitory activities (Table 2).

Collectively, the structural optimization and SAR studies led to the discovery of compound **25**, which exhibited high potency against SIRT2, better than the hit compound **20** and positive control AGK2. Subsequently, the IC_50_ value of **25** was then measured against SIRT2, and the IC_50_ curve has been presented in Figure 3. The study observed that compound **25** inhibited SIRT2 via a dose dependent manner with an IC_50_ value of 2.47 μM, which is more potent than AGK2 (with an IC_50_ value of 17.75 μM). Molecular docking was then used to investigate the possible binding mode of **25** with SIRT2. The results indicated that **25** appeared to fit well with the induced hydrophobic pocket (Figure 4) [44,45]. The carboxyl acid group of **25** is likely positioned to make hydrogen-bonding interactions with the main chain of Asp170 and the side chain of Thr171 and Tyr139. The furan and pyridine moiety likely have hydrophobic contacts with hydrophobic residues Phe119, Phe234, Phe131, Leu138, and Ile169 (Figure 4). Notably, the pyridine appears to form edge-to-face aromatic interactions with Phe119, and fits well with the pocket around Phe119, Phe131, and Phe234, suggesting that introducing substituents on pyridine may result in a clash with these three residues. Together, these docking results may explain why the replacement of the carboxyl acid group or the introduction of substituents on pyridine leads to a decrease in SIRT2 inhibition, and indicates the possible inhibition mode for this series of compounds.

## 3. Materials and Methods

### 3.1. Synthesis

As previously reported, proton (^1^H) and carbon (^13^C) NMR spectra were recorded on a Bruker AV-400 (Bruker Company, Billerica, Germany) instrument and are reported in ppm relative to tetramethylsilane (TMS) and referenced to the solvent in which the spectra were collected. Unless otherwise noted, all of the commercially available starting materials, reagents, and solvents and reagents were used without further purification. The analytical thin-layer chromatography (TLC) was run on Merck silica gel 60 F-254 (Qingdao Haiyang, Qingdao, China). The spots on the plates were visualized under UV light (λ = 254 nm). Purification was performed on silica gel chromatography with EtOAc—petroleum ether or CH_2_Cl_2_-MeOH solvent systems. The melting points were measured on an electrothermal melting point apparatus without correction (JIAHANG, Shanghai, China. ESI-MS was obtained on a Shimadzu-2010EV series liquid chromatograph mass spectrometer (Shimadzu, Tokyo, Japan). High-resolution mass spectra (HRMS) were determined using a SCIEX X500 QTOF mass spectrometer (Shanghai Sciex Analytical Instrument Trading Co., Shanghai, China). All target compounds were purified to >95% purity, as determined by the high-performance liquid chromatography (HPLC). The HPLC analysis was performed on a Waters 2695 HPLC system equipped with a Kromasil C18 column (4.6 mm × 250 mm, 5 μm, Waters, Milford, MA, USA).

#### 3.1.1. General Procedure for the Preparation of Key Intermediates **5a**–**5i**

A mixture of substituted iodobenzenes (**1a**–**1i**, 15 mmol), (5-formylfuran-2-yl)boronic acid (**2**, 15 mmol), bis(triphenylphosphine)palladium(II) chloride (Pd(Pph_3_)_2_Cl_2_, 0.6 mmol) and sodium carbonate (Na_2_CO_3_, 30 mmol) in MeCN/H_2_O (10 mL /10 mL) was stirred for 1 h at 60 °C. Upon completion of the reaction as determined by TLC, MeCN was removed by a rotary evaporator under reduced pressure, and the residue was acidated with 1M HCl solution (pH 7) and filtered. Next, the filtrate was partitioned between water (60 mL) and ethyl acetate (3 × 50 mL). The organic layer was dried over magnesium sulfate anhydrous (MgSO_4_), filtered and concentrated in vacuo. The crude products were purified by column chromatography with appropriate eluents to give the coupling products **3a**–**3i**, in 80–86% yields.

To a solution of the coupling products **3a**–**3i** (12 mmol) in EtOH (25 mL), hydroxylamine hydrochloride (NH_2_OH.HCl, 14.4 mmol) and sodium acetate (NaOAc, 14.4 mmol) were added and the mixture was stirred at reflux for 0.5 h. When TLC indicated that the reaction was finished, the reaction solution was concentrated and the residue was partitioned between water (50) and ethyl acetate (3 × 50 mL). The combined organic layer was dried over MgSO_4_, filtered and concentrated in vacuo to give the crude products **4a**–**4i**, which were used without further purification. Subsequently, to a stirring solution of condensation products, **4a**–**4i** (12 mmol) in EtOH (25 mL) was added to zinc powder (Zn, 12 mmol) and 3 M hydrochloric acid (HCl, 8.0 mL) at ambient temperatures. The reaction mixture was heated to 80 °C for further 2 h. After completion (monitored by TLC), the solvent was removed in vacuo, the crude residue was treated with 100 mL of ice water, and the pH was adjusted to 7–8 with saturated NaHCO_3_. Then, the mixture was filtered by diatomite and extracted with ethyl acetate (3 × 80 mL). The combined extracts were dried, concentrated and purified by column chromatography with appropriate eluents with three ethylamine (Et_3_N, TEA) to afford the desired intermediates **5a–5i** in high yields.

*1-((5-(2,5-Dichlorophenyl)furan-2-yl)methyl)-3-(quinolin-5-yl)urea* (**12**). A solution of quinolin-5-amine (**7**, 250 mg, 1.73 mmol) and TEA (200 μL, 2.03 mmol) dissolved in CH_2_Cl_2_ (DCM, 15 mL) was slowly dripped into a stirred solution of triphosgene (BTC, 256 mg, 0.85 mmol) in DCM (10 mL) by using a constant-pressure dropping funnel. Then, the mixture was stirred for another 0.5 h at room temperature (RT). After evaporation of the solvent, the residue was taken up in DCM (30 mL), and (5-(2,5-dichlorophenyl)furan-2-yl)methanamine (**5b**, 230 mg, 0.95 mmol) was added directly to the residue. The reaction mixture was stirred at RT for 6 h, and the solvent was subsequently removed in vacuo. The residue obtained was purified by column chromatography (*V*(PE):*V*(EA) = 1:1) to give the desired target compound **12** (343 mg, 0.84 mmol) in 88% yield. 96.8% HPLC purity. Mp: 245–246 °C. ^1^H-NMR (400 MHz, *DMSO-d*_6_) δ 8.92 (s, 1H), 8.89 (dd, *J* = 4.0 Hz, *J* = 4.0 Hz, 1H), 8.54 (d, *J* = 8.4 Hz, 1H), 8.06–8.04 (m, 1H), 7.86 (d, *J* = 2.8 Hz, 1H), 7.69 (s, 1H), 7.68 (d, *J* = 2.8 Hz, 1H), 7.60–7.53 (m, 2H), 7.39 (dd, *J* = 8.4 Hz, *J* = 2.4 Hz, 1H), 7.20 (d, *J* = 3.2 Hz, 2H), 6.53 (d, *J* = 3.2 Hz, 1H), 4.47 (d, *J* = 5.6 Hz, 2H) ppm. ^13^C-NMR (101 MHz, *DMSO-d*_6_) δ 155.9, 154.7, 150.7, 148.7, 147.3, 135.8, 133.0, 132.7, 130.8, 130.3, 129.9, 128.6, 127.7, 126.9, 123.8, 121.3, 121.0, 117.4, 113.6, 109.5, 36.90 ppm. HRMS: *m*/*z* calcd for C_21_H_16_N_3_O_2_ [M + H]^+^ 412.0577, found 412.0573.

*Ethyl 4-(5-((3-(pyridin-3-yl)ureido)methyl)furan-2-yl)benzoate* (**17**). The title compound was prepared from pyridin-3-amine (**9**) and ethyl 4-(5-(aminomethyl)furan-2-yl)benzoate (**5a**) using the same method as compound **12**, purified by column chromatography (*V*(DCM):*V*(MeOH) = 30:1). Yield: 82%. HPLC purity: 98.6%. Mp: 182–184 °C. ^1^H-NMR (400 MHz, *DMSO-d*_6_) δ 9.45 (s, 1H), 8.59 (d, *J* = 2.4 Hz, 1H), 8.12 (dd, *J* = 4.4, 1.2 Hz, 1H), 7.99 (d, *J* = 8.4 Hz, 2H), 7.93–7.88 (m, 1H), 7.82 (d, *J* = 8.4 Hz, 2H), 7.26 (dd, *J* = 8.4, *J* = 4.4 Hz, 1H), 7.18 (s, 1H), 7.10 (d, *J* = 3.2 Hz, 1H), 6.46 (d, *J* = 3.2 Hz, 1H), 4.40 (d, *J* = 5.6 Hz, 2H), 4.32 (q, *J* = 7.2 Hz, 2H), 1.33 (t, *J* = 7.2 Hz, 3H) ppm. ^13^C-NMR (101 MHz, *DMSO-d*_6_) δ 165.8, 155.6, 155.0, 151.5, 142.6, 139.9, 137.5, 134.8, 130.3, 128.5, 124.8, 124.0, 123.6, 114.6, 109.8, 61.2, 45.8, 31.2 ppm. HRMS: *m*/*z* calcd for C_20_H_20_N_3_O_4_ [M + H]^+^ 366.1448 found 366.1444.

*1-(Pyridin-3-yl)-3-((5-(4-(trifluoromethyl)phenyl)furan-2-yl)methyl)urea* (**18**). The title compound was prepared from pyridin-3-amine (**9**) and ((5-(4-(trifluoromethyl)phenyl)furan-2-yl)methyl)-l2-azane (**5h**) using the same method as compound **12**, purified by column chromatography (*V*(PE):*V*(EA) = 3:1). Yield: 83%. HPLC purity: 98.0%. Mp: 183–187 °C. ^1^H-NMR (400 MHz, *DMSO-d*_6_) δ 8.87 (s, 1H), 8.58 (s, 1H), 8.14 (s, 1H), 7.91 (d, *J* = 12.8 Hz, 2H), 7.87 (s, 1H), 7.77 (d, *J* = 8.0 Hz, 2H), 7.27 (dd, *J* = 12.8 Hz, *J* = 2.8 Hz, 1H), 7.11 (d, *J* = 3.2 Hz, 1H), 6.89 (t, *J* = 5.6 Hz, 1H), 6.46 (d, *J* = 3.2 Hz, 1H), 4.41 (d, *J* = 5.6 Hz, 2H) ppm. ^13^C-NMR (10 MHz, *DMSO-d*_6_) δ 155.5, 154.9, 154.1, 151.0, 143.6, 142.1, 139.3, 137.9, 134.3, 130.0, 126.4, 126.3, 125.5, 124.0, 109.7, 45.8 ppm. HRMS: *m*/*z* calcd for C_18_H_15_N_3_O_2_ [M + H]^+^ 362.1061, found 362.1070.

*4-(5-((3-(Quinolin-5-yl)ureido)methyl)furan-2-yl)benzoic Acid* (**20**). A mixture of ethyl 4-(5-((3-(quinolin-5-yl)ureido)methyl)furan-2-yl)benzoate (**11**, 415 mg, 1.0 mmol), which was prepared from quinolin-5-amine (**9**) and ethyl 4-(5-(aminomethyl)furan-2-yl)benzoate (**5a**) using the same method as compound **12**, and NaOH (127 mg, 3.0 mmol) reacted for 2 h in the solution of EtOH/H_2_O (10 mL/10 mL) at 80 °C. After evaporation of the organic solvent, the residue was treated with 50 mL of ice water, and the PH was adjusted to 6–7 with diluted HCl. Next, the mixture was extracted with ethyl acetate (3 × 50 mL). The combined extracts were washed with brine, dried, and concentrated. The residue obtained was purified by column chromatography (*V*(PE):*V*(EA) = 2:1) to give the final compound **20** (344 mg) in 89% yield. HPLC purity: 98.2%. Mp: 285–287 °C. ^1^H-NMR (400 MHz, *DMSO-d*_6_) δ 9.84 (s, 1H), 9.01 (d, *J* = 8.4 Hz, 1H), 8.86 (d, *J* = 4.0 Hz, 1H), 8.20 (t, *J* = 5.6 Hz, 1H), 8.14 (q, *J* = 5.6 Hz 1H), 7.94 (s, 1H), 7.92 (s, 1H), 7.68–7.62 (m, 4H), 7.50 (q, *J* = 4.0 Hz, 1H), 6.89 (d, *J* = 3.2 Hz, 1H), 6.45 (d, *J* = 3.2 Hz, 1H), 4.43 (d, *J* = 5.2 Hz, 1H) ppm. ^13^C-NMR (101 MHz, *DMSO-d*_6_) δ 170.8, 156.2, 153.7, 152.8, 150.5, 148.7, 136.5, 131.6, 130.1, 129.9, 123.0, 122.7, 121.0, 120.7, 116.3, 107.2, 101.7, 99.6, 36.9 ppm. HRMS: *m*/*z* calcd for C_22_H_18_N_3_O_4_ [M + H]^+^ 388.1252, found 388.1254.

*4-(5-((3-Phenylureido)methyl)furan-2-yl)benzoic Acid* (**21**). The title compound was prepared from ethyl 4-(5-((3-phenylureido)methyl)furan-2-yl)benzoate (**13**) using the same method as compound **20**, purified by column chromatography (*V*(PE):*V*(EA) = 2:1). Yield: 86%. HPLC purity: 97.6%. Mp: 285–288 °C. ^1^H-NMR (400 MHz, *DMSO-d*_6_) δ 9.28 (s, 1H), 7.99 (d, *J* = 8.0 Hz, 2H), 7.70 (d, *J* = 8.0 Hz, 2H), 7.85 (d, *J* = 8.0 Hz, 2H), 7.38 (s, 1H), 7.21 (t, *J* = 7.6 Hz, 2H), 6.95 (d, *J* = 3.2 Hz, 1H), 6.88 (t, *J* = 7.2 Hz, 1H), 6.40 (d, *J* = 2.8 Hz, 1H), 4.37 (d, *J* = 5.2 Hz, 2H) ppm. ^13^C-NMR (101 MHz, *DMSO-d*_6_) δ 168.9, 155.8, 154.6, 152.0, 141.2, 133.1, 130.3, 129.0, 123.1, 121.3, 118.0, 109.4, 108.5, 36.8 ppm. HRMS: *m*/*z* calcd for C_19_H_16_N_2_O_4_Na [M + Na]^+^ 359.0968, found 359.0965.

*4-(5-((3-Phenylthioureido)methyl)furan-2-yl)benzoic Acid* (**22**). The title compound was prepared from ethyl 4-(5-((3-phenylthioureido)methyl)furan-2-yl)benzoate (**14**) using the same method as compound **20**, purified by column chromatography (*V*(PE):*V*(EA) = 4:1). Yield: 87%. HPLC purity: 97.6%. Mp: 281–282 °C. ^1^H-NMR (400 MHz, *DMSO-d*_6_) δ 11.04 (s, 1H), 9.44 (s, 1H), 7.95 (d, *J* = 8.4 Hz, 3H), 7.70 (d, *J* = 8.4 Hz, 2H), 7.64 (d, *J* = 8.0 Hz, 2H), 7.30 (t, *J* = 4.0 Hz, 2H), 7.07 (d, *J* = 8.4 Hz, 1H), 6.96 (d, *J* = 3.2 Hz, 1H), 6.48 (d, *J* = 3.2 Hz, 1H), 4.82 (d, *J* = 5.6 Hz, 2H) ppm. ^13^C-NMR (101 MHz, *DMSO-d*_6_) δ 207.0, 181.4, 169.6 152.9, 152.4, 140.7, 132.7, 130.3, 128.7, 124.0, 123.1, 123.0, 110.1, 108.1, 31.2 ppm. HRMS: *m*/*z* calcd for C_19_H_17_N_2_O_3_S [M + H]^+^ 353.0954, found 353.0962.

*4-(5-((3-(4-Cyano-3-fluorophenyl)ureido)methyl)furan-2-yl)benzoic Acid* (**23**). The title compound was prepared from ethyl 4-(5-((3-(4-cyano-3-fluorophenyl)ureido)methyl)furan-2-yl)benzoate (**15**) using the same method as compound **20**, purified by column chromatography (*V*(DCM):*V*(MeOH) = 30:1). Yield: 92%. HPLC purity: 97.2%. Mp: 192–193 °C. ^1^H-NMR (400 MHz, *DMSO-d*_6_) δ 10.97 (s, 1H), 8.53 (s, 1H), 7.96 (d, *J* = 8.4 Hz, 2H), 7.83 (dd, *J* = 12.8 Hz, *J* = 2.0 Hz 1H), 7.70 (d, *J* = 8.4 Hz, 3H), 7.41 (d, *J* = 8.4 Hz, 1H), 6.93 (d, *J* = 3.2 Hz, 1H), 6.41 (d, *J* = 3.2 Hz, 1H), 4.40 (d, *J* = 5.6 Hz, 2H) ppm. ^13^C-NMR (101 MHz, *DMSO-d*_6_) δ 207.0, 169.6, 165.0, 162.5, 155.2, 153.6, 152.4, 148.45 134.3, 132.7, 130.3, 123.1, 115.4, 114.4, 109.6, 108.0, 45.9 ppm. HRMS: *m*/*z* calcd for C_20_H_15_FN_3_O_4_ [M + H]^+^ 380.1041, found 380.1040; C_20_H_14_FN_3_O_4_Na [M + Na]^+^ 402.0855, found 402.0857.

*2-Hydroxy-4-(5-((3-phenylureido)methyl)furan-2-yl)benzoic Acid* (**24**). The title compound was prepared from ethyl 2-hydroxy-4-(5-((3-phenylureido)methyl)furan-2-yl)benzoate (**16**) using the same method as compound **20**, purified by column chromatography(*V*(DCM):*V*(MeOH) = 30:1). Yield: 86%. HPLC purity: 96.6%. Mp: 224–227 °C. ^1^H-NMR (400 MHz, *DMSO-d*_6_) δ 12.05 (s br, 2H), 8.60 (s, 1H), 7.82 (d, *J* = 8.5 Hz, 1H), 7.42 (d, *J* = 8.0 Hz, 2H), 7.24 (d, *J* = 12.8 Hz, 4H), 7.10 (d, *J* = 3.2 Hz, 1H), 6.91 (s, 1H), 6.68 (s, 1H), 6.43 (d, *J* = 3.2 Hz, 1H), 4.38 (d, *J* = 5.6 Hz, 2H) ppm. ^13^C-NMR (101 MHz, *DMSO-d*_6_) δ 172.1, 162.1, 155.5, 155.2, 151.2, 140.8, 137.0, 131.5, 129.2, 121.7, 118.3, 114.6, 111.9, 111.1, 110.2, 109.6, 36.9 ppm. HRMS: *m*/*z* calcd for C_19_H_17_N_2_O_5_ [M + H]^+^ 353.1132, found 353.1140.

*4-(5-((3-(Pyridin-3-yl)ureido)methyl)furan-2-yl)benzoic Acid* (**25**). The title compound was prepared from ethyl 4-(5-((3-(pyridin-3-yl)ureido)methyl)furan-2-yl)benzoate (**17**) using the same method ascompound **20**, purified by column chromatography (*V*(PE):*V*(EA) = 3:1). Yield: 95%. HPLC purity: 98.6%. Mp: 218–220 °C. ^1^H-NMR (400 MHz, *DMSO-d*_6_) δ 12.98 (s br, 1H), 8.84 (s, 1H), 8.58 (s, 1H), 8.14 (d, *J* = 4.0 Hz, 1H), 7.99 (d, *J* = 8.4 Hz, 2H), 7.93 (d, *J* = 8.4 Hz, 1H), 7.79 (d, *J* = 8.0 Hz, 2H), 7.29 (q, *J* = 4.8 Hz, 1H), 7.08 (d, *J* = 3.2 Hz, 1H), 6.87 (t, *J* = 5.6 Hz, 1H), 6.45 (d, *J* = 3.2 Hz, 1H), 4.41 (d, *J* = 5.6 Hz, 2H) ppm. ^13^C-NMR (101 MHz, *DMSO-d*_6_) δ 167.5, 155.4, 154.7, 151.6, 142.7, 140.1, 137.5, 134.5, 130.5, 129.5, 125.2, 124.1, 123.5, 109.8, 109.5, 31.2 ppm. HRMS: *m*/*z* calcd for C_18_H_16_N_3_O_4_ [M + H]^+^ 338.1135, found338.1138.

*4-(5-((3-(5-Bromo-2-methylpyridin-3-yl)ureido)methyl)furan-2-yl)benzoic acid* (**26**). The title compound was prepared from ethyl 4-(5-((3-(5-bromo-2-methylpyridin-3-yl)ureido)methyl)furan-2-yl)benzoate (**19**) using the same method as compound **20**, purified by column chromatography (*V*(DCM):*V*(MeOH) = 10:1). Yield: 90%. HPLC purity: 97.6%. Mp: 250–254 °C. ^1^H-NMR (400 MHz, *DMSO-d*_6_) δ 12.96 (s, 1H), 8.56 (d, *J* = 2.8 Hz, 1H), 8.35 (s, 1H), 8.14 (d, *J* = 2.8 Hz, 1H), 7.98 (d, *J* = 8.4 Hz, 2H), 7.80 (d, *J* = 8.4 Hz, 2H), 7.58 (s, 1H), 7.09 (d, *J* = 3.2 Hz, 1H), 6.49 (d, *J* = 3.2 Hz, 1H), 4.42 (d, *J* = 5.6 Hz, 2H), 2.40 (s, 3H) ppm. ^13^C-NMR (101 MHz, *DMSO-d*_6_) δ 167.4, 155.4, 154.4, 151.8, 146.2, 142.0, 136.2, 134.4, 130.5, 129.6, 127.9, 123.5, 117.5, 110.0, 109.5, 36.8, 21.38 ppm. LCMS *m*/*z*: 428.0 [M − H]^−^.

*4-(5-((N-hydroxy-2-phenylacetamido)methyl)furan-2-yl)benzoic Acid Ethyl* (**30**). To a solution of (*E*)-4-(5-((hydroxyimino)methyl)furan-2-yl)benzoate (**4a**, 610 mg, 2.35 mmol) in methyl alcohol (10 mL), sodium cyanoborohydride (440 mg, 1.5 mmol) and 12 M hydrochloric acid (780 μL, 9.4 mmol) were added at 0 °C. Then, the mixture was stirred at room temperature for 4 h. When TLC indicated that the reaction was finished, the reaction solution was concentrated and the residue was basified with 6 N sodium hydroxide solution (pH 8) and extracted several times with ethyl acetate. The combined organic extracts were dried (Na_2_SO_4_), and concentrated under reduced pressure to yield the reduction product ethyl 4-(5-((hydroxyamino)methyl)furan-2-yl)benzoate (**27**, 328 mg) in 54% yield. Next, 2-phenylacetyl chloride (**28**, 195 mg, 1.27 mmol) and NaHCO_3_ (106 mg, 1.27 mmol) were added to the solution of **27** (328 mg, 1.27 mmol) in diethyl ether (15 mL) and the reaction was stirred at room temperature for 6 h. Upon completion of the reaction as determined by TLC, the resulting solution was concentrated under reduced pressure to dryness and the residue was purified by silica gel column chromatography to give the light yellow compound **29** in 78% yield. The target compound **30** was gained from ethyl 4-(5-((*N*-hydroxy-2-phenylacetamido)methyl)furan-2-yl)benzoate (**29**) using the same method as compound **20**, purified by column chromatography (*V*(PE):*V*(EA) = 3:1). Yield: 89%. HPLC purity: 97.0%. Mp: 162–163 °C. ^1^H-NMR (400 MHz, *DMSO-d*_6_) δ 12.95 (s, 1H), 10.15 (s, 1H), 7.98 (d, *J* = 8.8 Hz, 2H) 7.75 (d, *J* = 8.4 Hz, 2H), 7.32–7.21 (m, 5H), 7.08 (d, *J* = 3.2 Hz, 1H), 6.49 (d, *J* = 3.2 Hz, 1H), 4.80 (s, 2H), 3.79 (s, 2H) ppm. ^13^C-NMR (101 MHz, *DMSO-d*_6_) δ 171.9, 167.4, 152.0, 151.8, 136.1, 134.4, 130.5, 129.9, 129.6, 128.6, 126.8, 123.6, 111.7, 109.4, 45.07, 38.9 ppm. HRMS: *m*/*z* calcd for C_20_H_16_NO_5_ [M − H]^−^ 350.1034, found 350.1066.

#### 3.1.2. Hantzsch-Involved Reductive Amination Used for Compounds **31**–**34**

To a solution of substituted 5-phenylfuran-2-carbaldehydes (**3a** and **3i**, 1.5 mmol), different amines (1.8 mmol) and diethyl 2,6-dimethyl- 1,4-dihydropyridine-3,5-dicarboxylate (hantzschester, 1.8 mmol) in DCM (25 mL), catalytic amount of molecular sieve and trifluoroacetic acid were added at room temperature, and the reaction was warmed to 45 °C and reacted for 6–12 h. After completion (monitored by TLC), the reaction was filtered, and the crude residue was obtained by concentrating the filtrate in vacuo. Finally, the crude residue was purified by column chromatography to give the desired compounds **31**–**34** in high yields.

*Ethyl 4-(5-((Benzylamino)methyl)furan-2-yl)benzoate* (**32**). Yield: 56%. HPLC purity: 98.1%. Mp: 250–251 °C. ^1^H-NMR (400 MHz, *DMSO-d*_6_) δ 7.98 (d, *J* = 8.4 Hz, 2H), 7.81 (d, *J* = 8.4 Hz, 2H), 7.37–7.30 (m, 4H), 7.23 (t, *J* = 8.4 Hz, 1H), 7.08 (d, *J* = 3.2 Hz, 1H), 6.44 (d, *J* = 3.2 Hz, 1H), 4.31 (q, *J* = 7.2 Hz, 2H), 3.74 (s, 4H), 2.80 (s br, 1H), 1.33 (t, *J* = 7.2 Hz, 3H) ppm. ^13^C-NMR (101 MHz, *DMSO-d*_6_) δ 165.9, 156.4, 151.3, 140.9, 135.0, 130.3, 128.6, 128.5, 128.3, 127.1, 123.5, 110.0, 109.7, 61.2, 52.5, 45.3, 14.67 ppm. LCMS *m*/*z*: 335.2 [M + H]^+^.

*Ethyl (E)-4-(5-((2-nicotinoylhydrazono)methyl)furan-2-yl)benzoate* (**33**). Yield: 72%. HPLC purity: 97.5%. Mp: 201–204 °C. ^1^H-NMR (400 MHz, *DMSO-d*_6_) δ 12.10 (s, 1H), 9.09 (d, *J* = 1.2 Hz, 1H),8.79 (d, *J* = 3.6 Hz, 1H), 8.41 (s, 1H), 8.28 (dt, *J* = 8.0 Hz, 1H), 8.05 (d, *J* = 8.4 Hz, 2H), 7.94 (d, *J* = 8.4 Hz, 2H), 7.62–7.58 (m, 1H), 7.36 (d, *J* = 3.6 Hz, 1H), 7.16 (d, *J* =3.6 Hz, 1H), 4.34 (q, *J* = 7.2 Hz, 2H), 1.35 (t, *J* = 7.2 Hz, 3H) ppm. ^13^C-NMR (101 MHz, *DMSO-d*_6_) δ 165.7, 162.2, 154.2, 152.9, 150.4, 149.0, 138.2, 136.0, 133.9, 130.4, 129.5, 129.4, 124.5, 124.1, 117.3, 111.3, 61.3, 14.7 ppm. HRMS: *m*/*z* calcd for C_20_H_18_N_3_O_4_ [M + H]^+^ 364.1260, found 364.1264.

*1-(5-(4-Bromophenyl)furan-2-yl)-N-(pyridin-3-ylmethyl)methanamine* (**34**). Yield: 95%. HPLC purity: 97.8%. Mp: 146–150 °C. ^1^H-NMR (400 MHz, *DMSO-d*_6_) δ 8.53 (s, 1H), 8.44 (d, *J* = 3.6 Hz, 1H), 7.76 (d, *J* = 7.6 Hz, 1H), 7.64–7.58 (m, 5H), 7.35–7.32 (m, 1H), 6.93 (d, *J* = 3.2 Hz, 1H), 6.39 (d, *J* = 3.2 Hz, 1H), 3.75 (s, 2H), 3.72 (s, 2H), 1.23 (s, 1H) ppm. ^13^C-NMR (101 MHz, *DMSO-d*_6_) δ 155.1, 151.3, 149.9, 148.4, 136.3, 136.2, 132.2, 130.1, 125.6, 123.8, 120.4, 109.8, 107.8, 49.9, 45.4 ppm. HRMS: *m*/*z* calcd for C_17_H_15_BrNO_2_ [M + H]^+^ 342.0368, found 343.0397 and 345.0387.

*4-(5-((phenylamino)methyl)furan-2-yl)benzoic Acid* (**35**). The title compound was prepared from ethyl 4-(5-((phenylamino)methyl)furan-2-yl)benzoate (**31**) using the same method ascompound **20**, purified by column chromatography (*V*(PE):*V*(EA) = 2:1). Yield: 96%. HPLC purity: 98.3%. Mp: 207–210 °C. ^1^H-NMR (400 MHz, *DMSO-d*_6_) δ 12.96 (s, 1H), 7.98 (d, *J* = 7.6 Hz, 2H), 7.78 (d, *J* = 7.6 Hz, 2H), 7.09 (t, *J* = 7.6 Hz, 2H), 7.05 (d, *J* = 3.2 Hz, 1H), 6.70 (d, *J* = 7.6 Hz, 2H), 6.57 (t, *J* = 7.2 Hz, 1H), 6.47 (d, *J* = 3.2 Hz, 1H), 6.18 (s br, 1H), 4.34 (s, 2H) ppm. ^13^C-NMR (101 MHz, *DMSO-d*_6_) δ 167.4, 155.2, 151.5, 148.7, 134.6, 130.5, 129.4, 129.3, 123.4, 116.7, 112.9, 110.1, 109.4, 40.5 ppm. LCMS *m*/*z*: 294.1 [M + H]^+^.

*4-(5-((Benzylamino)methyl)furan-2-yl)benzoic Acid* (**36**). The title compound was prepared from ethyl 4-(5-((benzylamino)methyl)furan-2-yl)benzoate (**32**) using the same method as compound **20**, purified by column chromatography (*V*(PA):*V*(EA) = 2:1). Yield: 92%. HPLC purity: 98.0%. Mp: 248–250 °C. ^1^H-NMR (400 MHz, *DMSO-d*_6_) δ 10.96 (s br, 1H), 8.00 (d, *J* = 8.4 Hz, 2H), 7.88 (d, *J* = 8.4 Hz, 2H), 7.60 (d, *J* = 6.4 Hz, 2H), 7.44–7.38 (m, 3H), 7.17 (d, *J* = 3.2 Hz, 1H), 6.81 (d, *J* = 3.2 Hz, 1H), 4.27 (s, 2H), 4.19 (s, 2H) ppm. ^13^C-NMR (101 MHz, *DMSO-d*_6_) δ 167.4, 153.3, 147.3, 134.0, 132.5, 130.6, 130.5, 130.1, 129.3, 129.1, 124.0, 115.0, 109.5, 50.0, 42.6 ppm. HRMS: *m*/*z* calcd for C_19_H_17_NO_3_ [M + H]^+^ 308.1265, found 308.1260.

*(E)-4-(5-((2-nicotinoylhydrazono)methyl)furan-2-yl)benzoic Acid* (**37**). The title compound was prepared from ethyl (*E*)-4-(5-((2-nicotinoylhydrazono)methyl)furan-2-yl)benzoate (**33**) using the same method as compound **20**, purified by column chromatography (*V*(PE):*V*(EA) = 2:1). Yield: 92%. HPLC purity: 97.2%. Mp: 237–240 °C. ^1^H-NMR (400 MHz, *DMSO-d*_6_) δ 12.10 (s, 1H), 9.09 (s, 1H), 8.79 (d, *J* = 3.6 Hz, 1H), 8.41 (s, 1H), 8.28 (dt, *J* = 8.4 Hz, *J* = 2.0 Hz, 1H), 8.05 (d, *J* = 8.4 Hz, 2H), 7.93 (d, *J* = 8.4 Hz, 2H), 7.62–7.58 (m, 1H), 7.34 (d, *J* = 3.6 Hz, 1H), 7.16 (d, *J* = 3.6 Hz, 1H) ppm. ^13^C-NMR (101 MHz, *DMSO-d*_6_) δ 164.7, 162.2, 154.6, 152.8, 150.9, 149.1, 138.4, 137.2, 136.0, 130.5, 129.6, 129.5, 124.2, 124.1, 117.1, 110.7 ppm. HRMS: *m*/*z* calcd for C_18_H_14_N_3_O_4_ [M + H]^+^ 336.0975, found 336.0952.

*4-(5-(Phenylsulfonamidomethyl)furan-2-yl)benzoic Acid* (**39**). The intermediate **5a** (100 mg, 0.43 mmol) reacted with benzenesulfonyl chloride (90 mg, 0.05 mmol,) in the presence of Et_3_N (179 μL, 1.30 mmol) at room temperature, in DCM (15 mL). When TLC indicated that the reaction was finished, the reaction was concentrated in vacuo and the pH was adjusted to 7–8 with saturated NaHCO_3_. Then, the water solution was extracted with ethyl acetate (3 ×). The combined extracts were concentrated to give brown crude product ethyl 4-(5-(phenylsulfonamidomethyl)furan-2-yl)benzoate (**38**) in 91% yield, which was used to synthesize the target compound **39**, in 90% yield. HPLC purity: 97.5%. Mp: 222–223 °C. ^1^H-NMR (400 MHz, *DMSO-d*_6_) δ13.00 (s, 1H), 8.36 (d, *J* = 6.0 Hz, 1H), 8.00 (d, *J* = 8.4 Hz, 2H), 7.85 (d, *J* = 6.8 Hz, 2H), 7.69 (d, *J* = 8.4 Hz, 2H), 7.62–7.55 (m, 3H), 6.98 (d, *J* = 3.6 Hz, 1H), 6.38 (d, *J* = 3.2 Hz, 1H) 4.19 (d, *J* = 6.0 Hz, 2H) ppm. ^13^C-NMR (101 MHz, *DMSO-d*_6_) δ 167.4, 152.1, 152.0, 141.2, 134.2, 132.7, 130.4, 129.5, 129.5, 126.9, 123.6, 111.1, 109.1, 39.9 ppm. HRMS: *m*/*z* calcd for C_18_H_16_NO_5_S [M + H]^+^ 358.0671, found 358.0670.

*2-Phenyl-N-((5-(p-tolyl)furan-2-yl)methyl)acetamide* (**43**). (5-(*p*-tolyl)furan-2-yl)methanamine (**5c**, 378 mg, 1.49 mmol) reacted with 2-phenylacetic acid (200 mg, 1.47 mmol) in the presence of 1-hydroxybenzotriazole (HOBT, 214 mg, 1.47 mmol), 1-(3-dimethylaminopropyl)-3-ethylcarbodiimide hydrochloride (EDCI, 282 mg, 1.47 mmol), and *N*,*N*-diisopropylethylamine (DIPEA, 0.21 mL, 4.3 mmol) in DCM (20 mL). The mixture was stirred at room temperature for 12 h. Then, the mixture was concentrated and partitioned between water (60 mL) and ethyl acetate (3 × 60 mL). The organic layer was dried over MgSO_4_, filtered, concentrated and purified by column chromatography (*V*(PE):*V*(EA) = 6:1) to give the target compound **43** (327 mg) in 73% yield. HPLC purity: 99.2%. Mp: 193–194 °C. ^1^H-NMR (400 MHz, *DMSO-d*_6_) δ 8.60 (t, *J* = 5.6 Hz, 1H), 7.54 (d, *J* = 8.0 Hz, 2H), 7.30 (d, *J* = 4.4 Hz, 4H), 7.24–7.21 (m, 3H), 6.78 (d, *J* = 3.2 Hz, 1H), 6.30 (d, *J* = 3.2 Hz, 1H), 4.33 (d, *J* = 5.6 Hz, 2H), 3.48 (s, 2H), 2.32 (s, 2H) ppm. ^13^C-NMR (101 MHz, *DMSO-d*_6_) δ170.5, 152.8, 152.2, 137.1, 136.8, 129.9, 129.4, 128.7, 128.2, 126.8, 123.7, 109.4, 106.1, 42.7, 36.3, 21.3 ppm. HRMS: *m*/*z* calcd for C_20_H_19_NO_2_ Na [M + Na]^+^ 328.1231, found 328.1228.

*N-((5-(4-methoxyphenyl)furan-2-yl)methyl)-2-phenylacetamide* (**44**). The title compound was prepared from intermediate **5d** using the same method as compound **43**, purified by column chromatography (*V*(PA):*V*(EA) = 7:1). Yield: 86%. HPLC purity: 98.2%. Mp: 196–197 °C. ^1^H-NMR (400 MHz, *DMSO-d*_6_) δ 8.59 (t, *J* = 5.6 Hz, 1H), 7.58 (d, *J* = 8.8 Hz, 2H), 7.30 (d, *J* = 4.4 Hz, 4H), 7.25–7.21 (m, 1H), 6.99 (d, *J* = 8.8 Hz, 2H), 6.69 (d, *J* = 3.2 Hz, 1H), 6.28 (d, *J* = 3.2 Hz, 1H), 4.33 (d, *J* = 5.6 Hz, 2H), 3.79 (s, 3H), 3.49 (s, 2H) ppm. ^13^C-NMR (101 MHz, *DMSO-d*_6_) δ 170.5, 159.1, 152.8, 151.8, 136.8, 129.4, 128.7, 126.8, 125.2, 123.8, 114.8, 109.4, 105.1, 55.6, 42.7, 36.3 ppm. HRMS: *m*/*z* calcd for C_20_H_19_NO_2_ Na [M + Na]^+^ 344.1170, found 344.1175.

*N-((5-(3-methoxyphenyl)furan-2-yl)methyl)-2-phenylacetamide* (**45**). The title compound was prepared from intermediate **5e** using the same method as compound **43**, purified by column chromatography (*V*(PE):*V*(EA) = 7:1). Yield: 91%. HPLC purity: 97.8%. Mp: 182–184 °C. ^1^H-NMR (400 MHz, *DMSO-d*_6_) δ 8.61 (t, *J* = 5.6 Hz, 1H), 7.35–7.29 (m, 5H), 7.25–7.23 (m, 2H), 7.21–7.20 (m, 1H), 6.89 (d, *J* = 3.2 Hz, 1H), 6.86 (dd, *J* = 8.0 Hz, *J* = 3.2 Hz, 1H), 6.32 (d, *J* = 3.6 Hz, 1H), 4.34 (d, *J* = 5.6 Hz, 2H), 3.80 (s, 3H), 3.50 (s, 2H) ppm. ^13^C-NMR (101 MHz, *DMSO-d*_6_) δ 170.6, 160.1, 152.7, 152.5, 136.8, 132.1, 130.5, 129.4, 128.7, 126.8, 116.2, 113.4, 109.5, 109.1, 107.4, 55.6, 42.7, 36.3 ppm. HRMS: *m*/*z* calcd for C_20_H_19_NO_2_ Na [M + Na]^+^ 344.1170, found 344.1174.

*Methyl 3-(5-(benzamidomethyl)furan-2-yl)benzoate* (**46**). The title compound was prepared from intermediate **5f** and benzoic acid using the same method as compound **43**, purified by column chromatography (*V*(PE):*V*(EA) = 8:1). Yield: 90%. HPLC purity: 97.9%. Mp: 197–198 °C. ^1^H-NMR (400 MHz, *DMSO-d*_6_) δ 9.09 (t, *J* = 6.4 Hz, 1H), 8.22 (s, 1H), 7.95 (d, *J* = 8.0 Hz, 1H), 7.91 (d, *J* = 7.6 Hz, 2H), 7.85 (d, *J* = 7.6 Hz, 1H), 7.59–7.53 (m, 2H), 7.50–7.46 (m, 2H), 7.03 (d, *J* = 3.2 Hz, 1H), 6.44 (d, *J* = 3.2 Hz, 1H), 4.57 (q, *J* = 5.6 Hz, 2H), 3.89 (s, 3H) ppm. ^13^C-NMR (101 MHz, *DMSO-d*_6_) δ166.7, 166.5, 153.6, 151.4, 134.6, 131.8, 131.3, 130.8, 130.0, 128.8, 128.3, 128.2, 127.8, 123.8, 109.9, 108.3, 52.8, 36.8 ppm. HRMS: *m*/*z* calcd for C_20_H_17_NO_4_ [M + H]^+^ 336.1210, found 336.1211; C_20_H_16_NO_4_Na [M + Na]^+^ 358.1010, found 358.1015.

*Methyl 3-(5-((2-Phenylacetamido)methyl)furan-2-yl)benzoate* (**47**). The title compound was prepared from intermediate **5f** and phenylacetic acid using the same method as compound **43**, purified by column chromatography (*V*(PE):*V*(EA) = 8:1). Yield: 90%. HPLC purity: 97.2%. Mp: 183–184 °C. ^1^H-NMR (400 MHz, *DMSO-d*_6_) δ 8.66 (t, *J* = 5.6 Hz, 1H), 8.21 (s, 1H), 7.92 (d, *J* = 8.0 Hz, 1H), 7.86 (d, *J* = 8.0 Hz, 1H), 7.57 (t, *J* = 8.0 Hz, 1H), 7.30 (d, *J* = 4.4 Hz, 4H), 7.25–7.20 (m, 1H), 7.01 (d, *J* = 3.2 Hz, 1H), 6.36 (d, *J* = 3.2 Hz, 1H), 4.37 (d, *J* = 5.6 Hz, 2H), 3.89 (s, 3H), 3.49 (s, 2H) ppm. ^13^C-NMR (101 MHz, *DMSO-d*_6_) δ 170.6, 166.5, 153.4, 151.5, 136.8, 131.2, 130.8, 129.9, 129.4, 128.7, 128.3, 126.8, 123.8, 109.8, 108.2, 52.8, 42.7, 36.2 ppm. HRMS: *m*/*z* calcd for C_21_H_19_NO_4_Na [M + Na]^+^ 372.1171, found 372.1177.

*Ethyl 4-(5-(Nicotinamidomethyl)furan-2-yl)benzoate* (**48**). The title compound was prepared from intermediate **5a** and nicotinic acid using the same method as compound **43**, purified by column chromatography (*V*(PE):*V*(EA) = 6:1). Yield: 78%. HPLC purity: 97.8%. Mp: 182–183 °C. ^1^H-NMR (400 MHz, *DMSO-d*_6_) δ 9.30 (t, *J* = 2.0 Hz, 1H), 9.07 (d, *J* = 1.6 Hz, 1H), 8.73 (d, *J* = 6.0 Hz, 1H), 8.25 (t, *J* = 8.0 Hz, 1H), 8.00 (d, *J* = 8.4 Hz, 2H), 7.81 (d, *J* = 8.4 Hz, 2H), 7.53 (d, *J* = 6.4 Hz, 1H), 7.11 (d, *J* = 3.6 Hz, 1H), 6.52 (d, *J* = 3.2 Hz, 1H), 4.61 (d, *J* = 5.6 Hz, 2H), 4.31 (q, *J* = 7.2 Hz, 2H), 1.34 (t, *J* = 7.2 Hz, 3H) ppm. ^13^C-NMR (101 MHz, *DMSO-d*_6_) δ 165.8, 165.4, 153.9, 152.5, 151.6, 149.0, 135.6, 134.8, 130.3, 130.1, 128.6, 124.0, 123.6, 110.3, 109.8, 61.2, 36.8, 14.7 ppm. LCMS *m*/*z*: 351.1 [M + H]^+^.

*4-(5-(Benzamidomethyl)furan-2-yl)benzoic Acid* (**49**). Using the intermediate **5a** and benzoic acid, the title compound **49** was synthesized via condensation reaction (87% yield), and hydrolysis reaction (92% yield) in turn. HPLC purity: 97.0%. Mp: 183–186 °C. ^1^H-NMR (400 MHz, *DMSO-d*_6_) δ 9.09 (t, *J* = 5.6 Hz, 1H), 7.98 (d, *J* = 8.4 Hz, 2H), 7.90 (d, *J* = 1.6 Hz, 2H), 7.79 (d, *J* = 8.8 Hz, 2H), 7.77–7.49 (m, 3H), 7.08 (d, *J* = 2.4 Hz, 1H), 6.47 (d, *J* = 3.2 Hz, 1H), 4.58 (d, *J* = 5.6 Hz, 2H) ppm. ^13^C-NMR (101 MHz, *DMSO-d*_6_) δ 167.5, 166.7, 154.2, 151.6, 134.5, 134.5, 131.9, 130.5, 129.5, 128.8, 127.8, 123.5, 110.1, 109.5, 36.8 ppm. LCMS *m*/*z*: 320.1 [M − H]^−^.

*4-(5-((2-Phenylacetamido)methyl)furan-2-yl)benzoic Acid* (**50**). Using the intermediate **5a** and phenylacetic acid, the title compound **49** was synthesized via condensation reaction (82% yield), and hydrolysis reaction (95% yield) in turn. HPLC purity: 97.1%. Mp: 228–230 °C. ^1^H-NMR (400 MHz, *DMSO-d*_6_) δ 12.98 (s, 1H), 8.65 (t, *J* = 5.4 Hz, 1H), 7.98 (d, *J* = 8.8 Hz, 2H), 7.75 (d, *J* = 8.4 Hz, 2H), 7.30 (d, *J* = 4.4 Hz, 4H), 7.26–7.22 (m, 1H), 7.05 (d, *J* = 3.2 Hz, 1H), 6.38, (d, *J* = 3.2 Hz, 1H), 4.37 (d, *J* = 5.6 Hz, 2H), 3.49 (s, 2H) ppm. ^13^C-NMR (101 MHz, *DMSO-d*_6_) δ170.6, 167.4, 154.0, 151.7, 136.8, 134.5, 130.5, 129.5, 129.5, 128.7, 126.9, 123.5, 109.9, 109.4, 42.7, 36.3 ppm. HRMS: *m*/*z* calcd for C_20_H_17_NO_4_Na [M + Na]^+^ 358.1026, found 358.1015.

*3-(5-(Benzamidomethyl)furan-2-yl)benzoic Acid* (**51**). The title compound was prepared from compound **46** via hydrolysis reaction (90% yield), purified by column chromatography (*V*(PE):*V*(EA) = 3:1). HPLC purity: 97.0%. Mp: 190–191 °C. ^1^H-NMR (400 MHz, *DMSO-d*_6_) δ 13.15 (s br, 1H), 9.20 (t, *J* = 6.4 Hz, 1H), 8.22 (s, 1H), 7.94–7.91 (m, 3H), 7.83 (d, *J* = 7.6 Hz, 1H), 7.54 (t, *J* = 8.0 Hz, 2H), 7.47 (t, *J* = 8.0 Hz, 2H), 7.00 (d, *J* = 3.2 Hz, 1H), 6.42 (d, *J* = 3.2 Hz, 1H), 4.55 (d, *J* = 5.6 Hz, 2H) ppm. ^13^C-NMR (101 MHz, *DMSO-d*_6_) δ 167.5, 166.7, 153.5, 151.5, 134.5, 132.0, 131.8, 131.1, 129.8, 128.8, 128.4, 127.8, 124.1, 109.8, 108.1, 36.7 ppm. HRMS: *m*/*z* calcd for C_19_H_15_NO_4_ [M + Na]^+^ 344.0852, found 344.0855.

*3-(5-((2-Phenylacetamido)methyl)furan-2-yl)benzoic Acid* (**52**). The title compound was prepared from compound **47** via hydrolysis reaction (91% yield), purified by column chromatography (*V*(PE):*V*(EA) = 3:1). HPLC purity: 97.2%. Mp: 220–221 °C. ^1^H-NMR (400 MHz, *DMSO-d*_6_) δ 8.85 (t, *J* = 5.6 Hz, 1H), 8.27 (s, 1H), 7.91 (d, *J* = 8.0 Hz, 2H), 7.58 (t, *J* = 8.0 Hz, 1H), 7.36–7.34 (m, 4H), 7.29–7.25 (m, 1H), 7.02 (d, *J* = 3.2 Hz, 1H), 6.40 (d, *J* = 3.2 Hz, 1H), 4.41 (d, *J* = 5.6 Hz, 2H), 3.55 (s, 2H) ppm. ^13^C-NMR (101 MHz, *DMSO-d*_6_) δ 170.7, 167.8, 153.2, 151.8, 136.8, 131.0, 129.6, 129.5, 128.7, 128.4, 127.6, 126.8, 124.1, 109.7, 107.8, 42.7, 36.2 ppm. HRMS: *m*/*z* calcd for C_20_H_17_NO_4_Na [M + Na]^+^ 358.1011, found 358.105.

### 3.2. Inhibition Assays

This study tested the inhibitory activities of the synthesized compounds against recombinant human SIRT2 proteins using a fluorogenic substrate p2270(Ac-Glu-Thr-Asp-Lys(Dec)-AMC)-coupled trypsin assay. The assay buffer is 25 mM Tris–HCl pH 8.0, 150 mM NaCl, and 10% glycerol. The test compounds were added to 60 μL of reaction mixture containing SIRT2 enzymes (0.2 μM), and each compound was prepared in a 3-fold dilution series (300 μM–15 nM) with the final DMSO concentration < 1%. After incubation at 25 °C for 30 min, the reaction started by the addition of the substrate p2270 (10 mM) and NAD^+^(400 mM) at 25 °C. After 2 h, 50 μL 3~4 U/μL trypsin and 4 mM nicotinamide were added to terminate the reaction, followed by further incubation for 30 min at 25 °C. The fluorescence intensity was measured using a microplate reader (λ_ex_ = 380 nm, λ_em_ = 460 nm). All determinations were performed in triplicate. The IC_50_ values were obtained using GraphPad Prism software as described previously.

### 3.3. Molecular Docking Assays

All the docking simulations were performed using AutoDock Vina. The crystal structure of SIRT2 complexed with an *N*-(3-(phenoxymethyl)phenyl)acetamide derivative (**24a**) (PDB ID: 5YQO) and was used as the docking template. All the water and solvant molecules, as well as **24a** were removed, and clean protein structure coordinates were obtained. AutoDockTools was used to assign Gasteiger-Marsili charges to the protein structure model, and merge non-polar hydrogens onto their respective heavy atoms of the protein structure (saved as pdbqt format). The 3D coordinates of the compound structures were prepared using the Discovery Studio viewer, followed by assigning atom types and partial charges using AutoDockTools (saved as pdbqt format). The binding site was defined as a rectangular grid, with the grid center coordinates of [x, y, z = −13.5, −10.1, −18.4] and the grid size of [25, 25, 25], to encompass the entire binding site. The number of possible docking poses were set as 10, and the other docking parameters were set as default. The docking results were inspected using PyMOL.

## 4. Conclusions

In this study, a series of (5-phenylfuran-2-yl)methanamine derivatives were synthesized. The SAR analyses of these compounds with SIRT2 led to the identification of compound **25** with 99 ± 2% @ 100 μM and 90 ± 3 % @ 10 μM inhibition against SIRT2. Meanwhile, **25** likely possesses better water solubility (cLogP = 1.63 and cLogS = −3.63). The IC_50_ measurements revealed that **25** had considerable potency against SIRT2 with an IC_50_ value of 2.47 μM, which is more potent than AGK2. The molecular docking analyses indicated that **25** fits well with the induced hydrophobic pocket of SIRT2. This study will aid future investigations to discover new potent and selective SIRT2 inhibitors to provide potential treatments for relevant diseases.

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
