# Peer review of "Discovery of (5-Phenylfuran-2-yl)methanamine Derivatives as New Human Sirtuin 2 Inhibitors"

_molecules, 2019, doi:10.3390/molecules24152724_

Round 1

Reviewer 1 Report

The authors present a series of compounds and investigate their potential as sirtuin 2 inhibitors, initiating their series from a previously published lead (20). Out of the 27 compounds, only one (25) shows significant improvement from the lead (90% vs 35% inhibition at 10 µM).

The introduction is modest, and fails to mention the very potent macrocyclic Sirt2-inhibitor S2iL5 (Structure, 2014, 22, 345-352).

The chemistry is well conducted and described.

The assay part is modest but adequate. However, the authors fail to cite original literature, and instead cites their own work. Both synthesis and use in sirtuin 2 assays of the substrate used (here arbitrarily named p2270) was reported in J. Med. Chem. 2016, 59, 1021-1031. Please change the citation to include this instead of reference 39, which uses a different assay.

Author Response

Responses to the Editor and Reviewers:

First, we would like to thank the editor and the reviewers for their constructive comments. We have carefully addressed the reviewers’ concerns and made corrections. Detailed descriptions related to the revisions are given as follows:

Reviewer 1’s Comments, Point 1:

 “The authors present a series of compounds and investigate their potential as sirtuin 2 inhibitors, initiating their series from a previously published lead (20). Out of the 27 compounds, only one (25) shows significant improvement from the lead (90% vs 35% inhibition at 10 µM).

Answer: Thank the reviewer for this comment.

Reviewer 1’s Comments, Point 2:

The introduction is modest, and fails to mention the very potent macrocyclic Sirt2-inhibitor S2iL5 (Structure, 2014, 22, 345-352).

Answer: Thank the reviewer for this comment. According to the reviewer's suggestion, we have included the mentioned macrocyclic Sirt2-inhibitor S2iL5 and the reference (Structure, 2014, 22, 345-352) in the revised manuscript. Please see Page 1, Line 41 and Ref. 29 in the revised manuscript.

Reviewer 1’s Comments, Point 3:

The chemistry is well conducted and described.

Answer: Thank the reviewer for this comment.

Reviewer 1’s Comments, Point 4:

The assay part is modest but adequate. However, the authors fail to cite original literature, and instead cites their own work. Both synthesis and use in sirtuin 2 assays of the substrate used (here arbitrarily named p2270) was reported in J. Med. Chem. 2016, 59, 1021-1031. Please change the citation to include this instead of reference 39, which uses a different assay.

Answer: Thank the reviewer for this comment. According to the reviewer's opinion, we have included the mentioned reference (J. Med. Chem. 2016, 59, 1021-1031). Please see Ref 42 in the revised manuscript.

Reviewer 2’s (Academic Editor’s) Comments, Point 1:

I have checked the second version of the manuscript, but apart the movement of the conclusion paragraph the other very very small adjustments were not devoted to the implementation of the computational part that is exctly the same in the two versions.

Answer: Thank the reviewer for this comment. Appropriate descriptions about the implementation of the computational part have been included in the revised manuscript. Please see Pages 14-15 in the revised manuscript

Reviewer 2 Report

The manuscript by Wang et al. describes the identification and analogue development of a new class of NAD+-dependent protein deacylases.  The manuscript is reasonably well-written although still contains numerous grammatical errors and non-idiomatic usage.  The authors are encouraged to have a native English speaker edit the manuscript.  Most significant errors are listed below the review.   The chemistry section describes the synthesis of compounds in sufficient detail and with sufficient clarity.  The activity tables provide inhibition of SIRT2 at two fixed concentrations (100 and 10 microM).  This is not sufficient for an informed comparison of the potency of the compounds given the differences in the shape of dose-response curves as shown in Figure 3.  Having IC50 values for all compounds would be the most informative.  Lacking this, having IC50 values for 10 or so compounds with high potency at 10 microM in the enzymatic assay would be useful.  The most potent compound identified in this study (compound 25) is more potent that the reference compound used (AGK-2), but significantly weaker than several other commercially available SIRT2 inhibitors (e.g. SirReal-2).  It’s not clear why this compound wasn’t used for comparison.  Finally, it has become standard in the field to compare the inhibitory activity of a compound against a panel of related targets (i.e. SIRT1 and SIRT3). The authors only provide inhibition data for SIRT2.  The docking studies and interpretation of the SAR based on the proposed model are adequate. The manuscript will be of moderate interest to medicinal chemists involved in development of sirtuin inhibitors but unlikely to have significant impact.  

Typographical errors:

Line 11: Reads “SIRT2, a member of the sirtuin family, was considered as a main aspect contributing to the pathogenesis of cancers, neurodegenerative diseases …” should read “SIRT2, a member of the sirtuin family, was considered a promising drug target in cancer, neurodegenerative diseases …”

Line 52: Reads “Recently, we screened out in-house database using the fluorescence-based method …” should read “Recently, we screened out in-house compound collection using the fluorescence-based method …”

Author Response

Responses to the Editor

  First, we would like to thank the editor for their constructive comments. Following the editor’s suggestion, we have resubmitted the manuscript, which using the correct templates and the computational part of the revised manuscript has been improved. Please see the revised manuscript.